# Belching in Gastroesophageal Reflux Disease: Literature Review

**DOI:** 10.3390/jcm9103360

**Published:** 2020-10-20

**Authors:** Akinari Sawada, Yasuhiro Fujiwara, Daniel Sifrim

**Affiliations:** 1Department of Gastroenterology, Osaka City University Graduate School of Medicine, Osaka 545-8585, Japan; m1164972@med.osaka-cu.ac.jp; 2Wingate Institute of Neurogastroenterology, Blizard Institute, Barts and The London School of Medicine and Dentistry, Queen Mary University of London, London E1 2AJ, UK; d.sifrim@qmul.ac.uk

**Keywords:** gastric belching, supragastric belching, epidemiology, impedance-pH monitoring, gastroesophageal reflux disease, non-erosive reflux disease, reflux hypersensitivity, functional heartburn

## Abstract

Belching is a common phenomenon. However, it becomes bothersome if excessive. Impedance–pH monitoring can classify the belching into two types: gastric belching and supragastric belching (SGB). The former is a physiological mechanism to vent swallowed air from the stomach, whereas the latter is a behavioral disorder. Gastroesophageal reflux disease (GERD) is the most relevant condition in both types of belching. Recent findings have raised awareness that excessive SGB possibly sheds light on the pathogenesis of a part of proton pump inhibitor (PPI) refractoriness in GERD. SGB could cause typical reflux symptoms such as heartburn, regurgitation or chest pain in two ways: SGB-induced gastroesophageal reflux or SGB-induced esophageal distension. In PPI-refractory GERD, it is important to detect hidden SGB as a cause of reflux symptoms since SGB requires psychological treatment instead of high dose PPIs or pain modulators. In the case of PPI-refractory GERD with excessive SGB, recent studies imply that the combination of a psychological approach and conventional treatment can improve treatment outcome.

## 1. Introduction

Belching is defined as “an audible escape of air from the esophagus or the stomach into the pharynx” [1]. Although everyone belches, it can be bothersome when excessive and/or triggering reflux symptoms. If patients cannot control belching in public, they often feel embarrassed, which profoundly disturbs their social lives. Such patients understandably seek medical care for their belching symptom. On the other hand, some patients predominantly complain of typical reflux symptoms rather than belching, even though excessive belching itself causes reflux symptoms [2,3,4]. Due to poor response to proton pump inhibitors (PPIs) therapy, those patients are often referred to specialists as PPI-refractory gastroesophageal reflux disease (GERD) [2,4,5,6]. Recent studies suggest that hidden SGB is a possible cause of PPI refractoriness in a deceptively large proportion of GERD. SGB has to be diagnosed appropriately as it requires dedicated psychological therapy. This literature review summarizes belching with regard to mechanisms, epidemiology, relevant conditions, and treatment, focusing on the association with GERD.

## 2. Two Types of Belching: Gastric Belching and Supragastric Belching

### 2.1. Gastric Belching (GB)

GB is a physiological mechanism to vent swallowed air from the stomach. Air-induced distension of the proximal stomach triggers transient lower esophageal relaxation (TLESR), which allows the gas to be released retrogradely to the pharynx (Figure 1 and Figure 2). TLESR is a vagal reflex via the brainstem, not a local reflex [7]. In detail, the vagal afferent conveys the stretch of cardiac stomach to the brainstem via nodose ganglion and solitary nucleus. Subsequently, vagal efferent projecting to LES from the brainstem releases nitric oxide and vasoactive intestinal peptide, which relaxes the LES [8]. During TLESR, esophageal shortening often occurs by much stronger contraction of the longitudinal muscle than circular muscle. The discordance of these two muscle layers stretches myenteric neurons to release nitric oxide, which is possibly involved with the mechanism of TLESR [9].

### 2.2. Supragastric Belching (SGB)

SGB is a behavioral disorder where a patient sucks or swallows air from the mouth into the esophagus, immediately followed by expelling it through the pharynx (Figure 1 and Figure 2). Regarding its physiological mechanism, the diaphragm contracts to generate negative pressure in the esophagus (thoracic cavity). The upper esophageal sphincter (UES) relaxes, letting air into the esophagus (i.e., suck) from the pharynx due to the pressure gradient. Afterwards, the air is emitted retrogradely into the pharynx, increasing gastric and esophageal pressure by abdominal straining [10]. Apart from air sucker type, a minority of patients use the pharyngeal pump (i.e., swallow) instead for the intake of air.

A small number of SGB and GB can be seen in healthy asymptomatic subjects. [11,12]. SGB becomes bothersome when excessive as it can impair quality of life [13] and induce pathological gastroesophageal reflux [3,6,14]. We previously proposed ≤13 episodes/24 h as the normal value of SGB [11]. However, it should be taken into consideration that personal factors can alter the threshold for feeling belching as irritating. In many cases, patients can identify warning symptoms such as throat/chest/abdominal discomfort to trigger SGB [3]. Presumably, patients learn the behavior to try to ease such an unpleasant sensation in some way, and with time it turns into a subconscious act. We sometimes encounter patients who remember the exact moment or time when they began having excessive SGB. In such cases, stressful life events might precipitate SBG. In animal experiments, Lang et al. found that esophageal distension can trigger a reflex of inhaling air followed by belching via vagal nerves and esophageal tension/mucosal mechanoreceptors [15]. In light of this finding, a limited part of SGB might be a reflex rather than a behavior, especially when gas reflux causes the initial esophageal distension.

Psychological factors can influence the frequency of SGB. Bredenoord et al. showed that patients have more SGBs when aware of impedance recording, whilst distraction by filling in questionnaires or speaking can reduce the number of SBGs [16]. Additionally, SGB is hardly observed during sleeping [17].

Regarding personality, our previous study found that patients with excessive SGB showed lower levels of neuroticism (i.e., tend to less worry about symptoms) and the same level of hypervigilance (i.e., tend not to be too alert for bodily sensation) compared to healthy subjects [3].

## 3. Conceptual Change of Belching Disorders and Aerophagia in Rome Diagnostic Criteria

Excessive belching was initially thought to be a consequence of aerophagia (frequent air swallowing) [18]. Therefore, Rome II defined excessive belching as a part of “Aerophagia”. The advent of intraluminal impedance monitoring enabled us to assess both the type and movement of bolus in the esophagus (i.e., gas or liquid, and antegrade or retrograde movement) [19,20]. With this technique, Bredenoord et al. found a novel type of belching named supragastric belching (SGB) [21]. This finding brought research about belching into the next level because, until this study, all the belching had been considered as gastric belching. Consequently, Rome III expanded the disease concept by adding “unspecified excessive belching (i.e., excessive belching without air swallowing)” as the other subcategory apart from “aerophagia”, and renamed the disorder as “Belching Disorders” [22]. At that time, the two terms, “aerophagia” and “SGB”, were used in a bewildering way since SGB was thought to be found only in “aerophagia” [21]. However, the two disorders have been segregated from each other recently because they have distinctive features or physiology. In aerophagia, esophageal peristalsis propels swallowed air, which ends up accumulating in the intestine and colon. It mainly manifests itself as bloating, abdominal distention and constipation, although some patients complain of PPI refractory GERD symptoms [4]. On the other hand, SGB does not involve any esophageal peristalsis, and swallowed air does not reach the stomach [23,24]. Nevertheless, aerophagia still lacks clear diagnostic criteria to distinguish from normal air swallowing. The latest Rome IV removed the term “Aerophagia” from “Belching disorders” in which belching was simply classified into excessive GB and SGB on the basis of the origin of the gas [1] (Figure 3).

## 4. Epidemiology

There are several studies reporting the prevalence of belching symptom in the general population, which ranges from 6.7% to 28.8% [25,26,27,28]. Subjects are more likely to complain of belching symptom when having concomitant typical reflux symptoms (i.e., heartburn and/or regurgitation) (Table 1). In GERD patients, the prevalence of belching ranges widely from 4.1–75.6% [14,29,30,31,32,33]. Interestingly, Klauser et al. [29] showed no difference in the prevalence of belching between normal and pathological acid exposure in GERD. It is important to note that these epidemiological studies include both types of belching (i.e., GB and SGB) due to interview-based survey. Besides, heterogeneous definitions of excessive belching and GERD can influence the results significantly. 

As for the prevalence of SGB, our previous studies found that 3.4% of patients referred to a tertiary GI physiology unit had excessive SGB [11], and the prevalence in PPI-refractory GERD patients differs between regions as the Japanese (18.5%) had lower prevalence of excessive SGB compared to the British (36.1%) [12].

## 5. Belching and Gastroesophageal Reflux Disease

Figure 4 illustrates the relationship between belching and GERD and other relevant conditions.

### 5.1. Gastric Belching (GB)

Twenty-four-hour impedance–pH monitoring showed more frequent air swallowing and GBs in GERD compared to normal subjects, where the number of GBs moderately correlates to the number of air swallowing [34]. Bravi et al. found that patients with PPI-refractory GERD have much more prandial aerophagia and postprandial GB than PPI responders [4]. However, GB might not cause liquid reflux despite sharing the common reflux mechanism (i.e., TLESR) [34,35], as gas reflux precedes liquid reflux in only up to half of acid mixed (i.e., gas and liquid) reflux [20,36]. Its causal link remains to be elucidated.

### 5.2. Supragastric Belching (SGB)

SGB can induce reflux, the mechanism of which is unclear (Figure 1B). There are three possibilities as follows: (1) SGB-induced esophageal distension might provoke A TLESR [37,38], (2) AN INCREASE IN abdominal pressure might bring up the gastric contents through the LES, (3) TLESR might elicit a reflex of UES relaxation, which IN TURN lets the air into the esophagus [39]. Glasinovic et al. showed that 26% of total acid exposure time is caused by SGB-induced reflux in GERD patients with excessive SGB [3]. This study also showed improvement of acid exposure by the reduction of SGB, which indicates that SGB itself elicits reflux and does not coincidentally occur close to reflux.

### 5.3. Impact of Supragastric Belching (SGB) on PPI-Refractory GERD

Recent studies have increasingly raised awareness of potential involvement of SGB on reflux symptoms (not belching symptom) in a fairly large proportion of PPI-refractory GERD. Yadlapati et al. [5] showed as much as 42% of PPI-refractory GERD patients have pathological SGB. Our study [2] found that prevalence varies between reflux phenotypes with 37.7% in non-erosive reflux disease (NERD) (i.e., acid exposure time > 6%), 39.7% in reflux hypersensitivity (RH) (i.e., acid exposure time < 4% and positive symptom reflux association) and 22% in functional heartburn (FH) (i.e., acid exposure time < 4% and negative symptom reflux association). More importantly, SGB is potentially responsible for approximately 40% of typical reflux symptoms (heartburn, regurgitation or chest pain) in RH [2], and the severity of reflux symptoms correlates to the extent of SGB positively [14]. These studies imply that treatment for SGB is required to relieve reflux symptoms in a part of PPI-refractory GERD.

Presumably, SGB provokes typical reflux symptoms in two ways: (1) inducing reflux and (2) air-induced esophageal distension (Figure 1D). Takeda et al. found that esophageal balloon distension triggers heartburn, and the likelihood of triggering the symptom depends on the extent of distension [40]. The ethnic/racial difference can influence the perception of SGB, as SGB is less associated with reflux symptoms in Japanese than in British [12].

### 5.4. Belching after Anti-Reflux Surgery

#### 5.4.1. Laparoscopic Fundoplication

Laparoscopic Nissen fundoplication (LNF) is one of the most common procedures for GERD, which creates a 360° wrap of the stomach around the esophagogastric junction (EGJ) and repairs the crura if necessary. By tightening the EGJ, LNF eliminates reflux as well as GBs. However, LNF makes patients feel difficult to belch and can produce other gas-related symptoms including bloating or flatulence [41,42,43]. These discomforts presumably lead the patients to perform SGB as a futile attempt to belch and end up increasing the number of SGBs postoperatively [44]. 

Regarding laparoscopic partial fundoplication (LPF), which constructs a looser wrap (e.g., 270° posterior wrap (Toupet)), it can reduce gastroesophageal reflux similar to LNF and lessen postoperative gas-related symptoms, with the ability to belch remaining. However, LPF also increases the number of SGBs [45,46]. 

#### 5.4.2. Magnetic Sphincter Augmentation Device (LINX Reflux Management System)

Magnetic sphincter augmentation device (MSA) is made up of titanium beads with a magnetic core in a ring shape, which is laparoscopically placed around the LES. MSA aims to reinforce the weak LES to resist the gastric pressure (i.e., gastroesophageal reflux) without interfering with the passage of swallowed food bolus. Previous meta-analyses showed MSA is superior to LNF concerning preservation of the ability to belch, with a similar extent of reflux symptom improvement [47,48]. Several case series also reported that almost all patients kept the ability to belch postoperatively [49,50,51]. However, there are few studies to objectively assess influence of MSA on belching using impedance monitoring.

## 6. Belching and Other Relevant Conditions

### 6.1. Functional Dyspepsia

Up to 80% of functional dyspepsia (FD) patients complain of belching symptom [33,52,53,54]. According to a study by Conchillo et al., a high prevalence of belching symptom in FD is probably attributed to more frequent air swallowing compared to healthy subjects [55]. Belching symptom was associated with lower quality of life and more weight loss in patients with FD [52]. It could be explained by higher sensitivity to gastric dilatation in FD patients with concomitant belching symptom than without [53]. However, it remains uncertain if epigastric discomfort causes air swallowing or vice versa. The two factors might interact mutually to increase distressing symptoms. Interestingly, FD shows less correlation between belching symptom and acid reflux than GERD, and PPIs do not improve belching in FD unlike that in GERD [33].

Although little is known about the association between FD and SGB, patients with excessive SGB often identify abdominal discomfort as a warning signal [3].

### 6.2. Globus

Globus is a functional gastrointestinal disorder consisting of a persistent or intermittent lump sensation in the throat [56]. Patients with globus had higher prevalence of pathological SGB and aerophagia than GERD patients [57]. There might be a causal link between SGB and globus as some patients recognize throat discomfort as a warning signal for SGB [58].

### 6.3. Bariatric Surgery (Sleeve Gastrectomy)

Sleeve gastrectomy is one of the bariatric surgeries, which removes a generous portion of the stomach on the greater curvature side to create a tubular gastric pouch. Burgerhart et al. showed laparoscopic sleeve gastrectomy (LSG) almost doubled the number of GBs (from 29.7 to 59.5/24 h) despite the decrease of liquid and air swallows, whereas the number of SGBs did not alter [59]. As LSG is known to increase esophageal acid exposure with the decrease of LES pressure [60], LSG might increase GB by the anatomical change of the stomach (i.e., lack of gastric fundus and small accommodation capacity) and the weakened LES.

### 6.4. Miscellaneous Conditions

Various disorders causing distressing symptoms around the chest and abdomen (i.e., peptic ulcer disease, pancreatitis, angina pectoris and symptomatic cholelithiasis) are reported to manifest in belching symptom [61]. However, their causal link remains vague.

## 7. Clinical Approach and Treatment for Belching

### 7.1. Clinical Approach to Excessive Belching

Patients with troublesome belching can be divided into two groups: GB- or SGB-predominant type. Li et al. found that both groups have similar reflux profile and air swallow, but the SGB-predominant type showed more belching episodes than GB-predominant type [62]. With respect to the symptom, Kessing et al. found the severity of belching symptom depends on the number of SGBs, not GBs [14]. The number or severity of belching could provide clinicians with a hint on the type of belching (i.e., SGB tends to be repetitive whereas GB tend to be single isolated events). However, impedance–pH monitoring, a gold standard modality, is often necessary for precise diagnosis. 

### 7.2. Clinical Approach to Hidden SGB in PPI-Refractory GERD

In PPI-refractory GERD, many patients do not report belching symptom even when SGB is a culprit of reflux symptoms. Therefore, clinicians have to pay attention to impedance tracing so that hidden SGB is not overlooked. As approximately 50–60% of the SGBs exist close to reflux episodes [2,6], manual editing of automatic analysis of all reflux episodes can identify pathologic SGB. It is important to evaluate to what extent SGB causes acid exposure or reflux symptom.

### 7.3. Treatment for Gastric Belching

Treatment for GB should target air swallowing and/or TLESR.

To suppress air swallowing, it is worth trying to provide instruction to eat slowly and avoid carbonated drinks. Several studies showed that PPI therapy improves belching symptom [30,63,64,65], except for one study reporting no benefit [66]. The effect of PPIs on reflux symptoms might stop air swallowing as a response to chest/abdominal discomfort. Anti-reflux surgery reduces the number of TLESRs [44]. Besides, gamma aminobutyric acid receptor type B (GABA_B_) receptor agonist, baclofen, can alleviate belching symptom by reducing the number of TLESRs [67] and increasing LES pressure [68]. However, baclofen has a side effect of drowsiness, and thus, is not widely used. Lesogaberan, peripherally acting as a GABA_B_ receptor agonist, showed its ability to reduce TLESR with little central nervous system effect [69]. Nevertheless, it has never been launched.

In most cases, patients suffer from both excessive GB and reflux symptom. In addition to the eating habit modification, the therapeutic options can be optimized on the basis of reflux phenotype. For patients with severe erosive reflux disease or NERD, laparoscopic fundoplication or a maximum dose of PPI plus baclofen would be the first choice. For RH and FH patients (normal acid exposure), adding baclofen to their mainstream therapy, which is pain modulators, is likely to be beneficial as long as patients can tolerate its side effect.

In functional dyspepsia, several drugs shows a suppressive effect on belching symptom, such as rifaximin [70], acotiamide (acetylcholinesterase inhibitor) [71,72], rebamipide (mucosal protective agents), and famotidine (H_2_ receptor antagonist) [73]. 

### 7.4. Treatment for Supragastric Belching

SGB benefits from psychological therapy to unlearn or interrupt the acquired behavioral. Dedicated cognitive behavior therapy [3,74,75] or speech therapy [76,77] have established its efficacy on SGB. Speech therapy improves the severity of belching symptom in 83% of SGB patients, whereas CBT reduces the number of SGBs objectively by >50% in half of patients [3]. These two treatments share the common concept consisting of cognitive and behavioral components. 

For instance, our CBT has five sessions. In the cognitive part, firstly therapists explain the mechanism of SGB so that patients can understand SGB is a subconscious but deliberate behavior. We should be careful not to make patients feel blamed in this process. Otherwise, it will impact adherence to the treatment negatively. Secondly, a warning signal (premonitory sensations) should be identified, which causes patients to commence SGB to deal with the sensation. In most cases, throat, chest, or abdominal discomfort are identified as a warning signal, which can be used for a cue to start exercise to stop SGB as below. Additionally, the thought that SGB is useful to ease that discomfort needs to be corrected. 

In the behavioral part, therapists instruct patients in the use of (1) diaphragmatic breathing, and (2) mouth opening/tongue position to make it physically impossible to perform SGB. Diaphragmatic breathing inhales and exhales, spending 3 s in each phase by moving the abdominal wall, not the thoracic wall. Patients need to breathe through the mouth with opening moderately and place the tongue behind the top front teeth. These procedures should be trained at least twice per day for 3–5 min in a supine or sitting position. Once patients get used to the breathing technique, they are encouraged to use it as often as possible, especially when noticing a warning signal or commencing a bunch of SGBs. Our previous study found a lower number of SGBs, lower hypervigilance, and higher CBT proficiency (i.e., possible to identify a warning signal, better acceptance and adherence to CBT), which are predictive factors for better outcome, and the therapeutic effect of CBT sustains for at least 12 months [58]. 

Concerning medications, a case has been reported that a combination of baclofen and gabapentin alleviates SGB [78]. On the other hand, a randomized controlled trial shows baclofen does not decrease the number of SGBs [79].

### 7.5. Treatment Implication for SGB With PPI-Refractory GERD

In this setting, SGB is possibly required for improving typical reflux symptoms along with the current mainstream therapy for PPI refractoriness [2,5]. On the basis of relevant literatures, we here propose a possible therapeutic approach for each reflux phenotype with excessive SGB.

#### 7.5.1. Excessive SGB with Severe Erosive Esophagitis or NERD

Sole CBT probably cannot reduce acid exposure to the physiological range in this scenario, although it depends to what extent SGB-induced reflux accounts for acid exposure time. In fact, a previous study showed CBT decreases acid exposure time significantly, however, it was inadequate to normalize it [3]. Therefore, it would be reasonable to perform ARS first to protect damaged esophageal mucosa and subsequently add CBT for SGB, as ARS might further increase the number of SGBs [44].

#### 7.5.2. Excessive SGB with RH

Excessive SGB possibly accounts for approximately 40% of typical reflux symptoms [2]. Considering SGB hardly responds to pain modulators or PPIs, dual therapy (i.e., pain modulators and CBT) might be beneficial for easing reflux symptoms. However, further study is warranted to see which treatment is the best, sole CBT, pain modulators or dual therapy.

#### 7.5.3. Excessive SGB with FH

FH has a lower prevalence of excessive SGB than RH [2], however, there is a possibility that SGB-induced distension causes reflux symptoms in FH like RH. However, there is no study to assess the association between SGB and reflux symptoms in FH. If that is the case, dual therapy of CBT and pain modulators might be effective.

## 8. Conclusions

Belching can be classified into GB and SGB by impedance monitoring. SGB causes not only belching symptom but reflux symptoms by esophageal distension. As recent studies stress a role of SGB on PPI refractoriness in GERD, SGB should be carefully detected even when patients do not complain of belching symptom much.

The therapeutic options for GB and SGB are still limited as there is little treatment for air swallowing, and no alternative to psychological treatment for SGB although up to 50% of patients do not respond to CBT. Further study is required to tackle these problems.

## Figures and Tables

**Figure 1 jcm-09-03360-f001:**
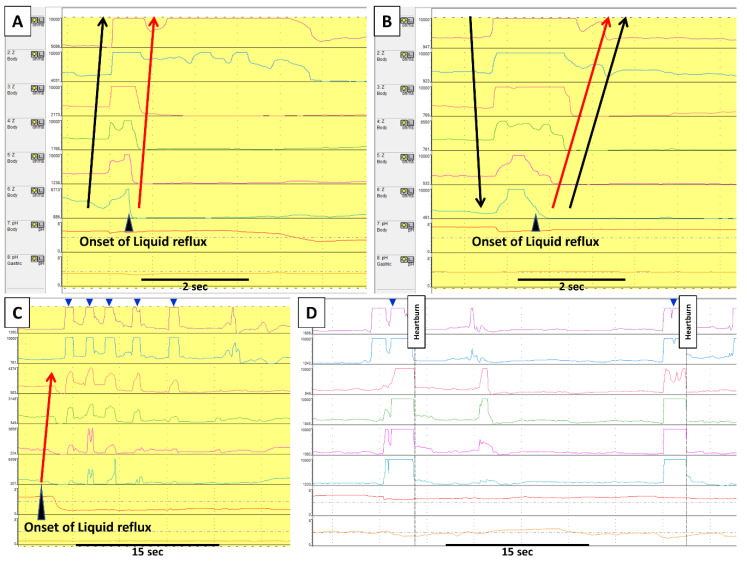
Gastric belching and supragastric belching in impedance–pH monitoring. Each row in impedance–pH tracings measures, from the bottom, gastric pH; esophageal pH at 5 cm above the lower esophageal sphincter (LES); and impedance at 3, 5, 7, 9, 15, and 17cm above the LES. In GB (**A**), an increase of impedance, indicating air from the stomach, moves retrogradely from the distal to the proximal esophagus. This GB is immediately followed by a liquid reflux episode (i.e., retrograde impedance decrease). On the other hand, SGB (**B**) starts with antegrade movement of impedance increase by sucking or swallowing air from the pharynx; subsequently, such increase returns to baseline retrogradely by abdominal straining. This SGB induces a liquid reflux episode. Apart from the SGB inducing reflux, there are two other patterns of the relationship between SGB and reflux. (**C**) shows that several SGBs (blue arrowheads) occur during liquid reflux in which an uncomfortable sensation by reflux might lead a patient to perform SGBs. (**D**) shows SGBs (blue arrowheads) without reflux. In this case, a patient constantly marked heartburn immediately after SGBs, which suggests the SGBs caused reflux symptoms by esophageal distension. Black arrows indicate the movement of air, whereas red arrows indicate liquid reflux.

**Figure 2 jcm-09-03360-f002:**
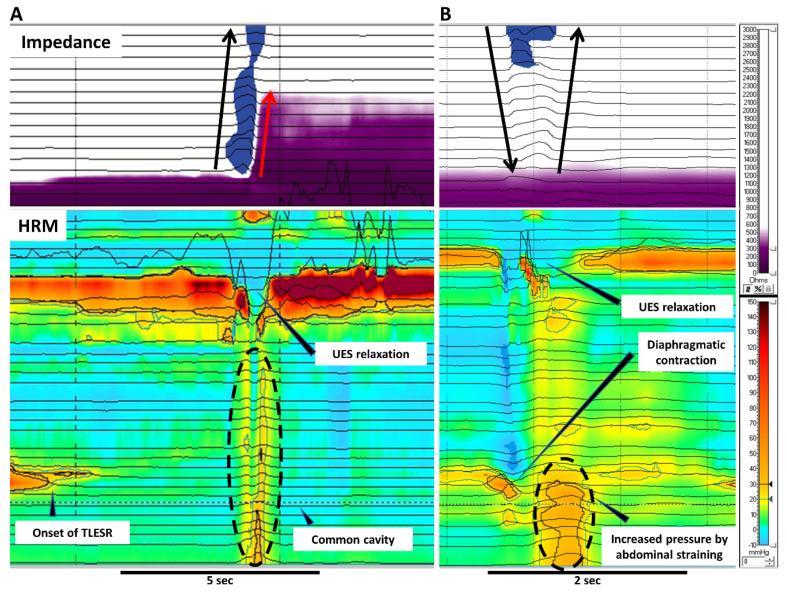
Gastric belching and supragastric belching in HRM combined with impedance. These pictures illustrate GB (**A**) and SGB (**B**) in simultaneous recording of impedance and HRM. (**A**) A GB occurs with liquid reflux during TLESR. GB can be recognized as a common cavity in HRM. The air evacuates from the stomach to the pharynx through the relaxed UES. (**B**) In SGB, the contraction of the diaphragm generates negative pressure in the esophagus. UES relaxation lets air into the esophagus followed by expelling the air by abdominal straining. No peristalsis is involved with the air movement in SGB. Black arrows indicate the movement of air whereas red arrows indicate liquid reflux. HRM; high resolution manometry, TLESR; transient lower esophageal sphincter relaxation, UES; upper esophageal sphincter.

**Figure 3 jcm-09-03360-f003:**
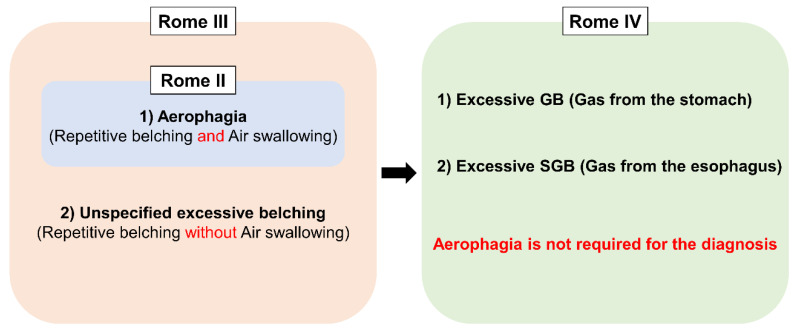
Conceptual change of Belching disorders and Aerophagia in Rome diagnostic criteria. GB; gastric belching, SGB; supragastric belching.

**Figure 4 jcm-09-03360-f004:**
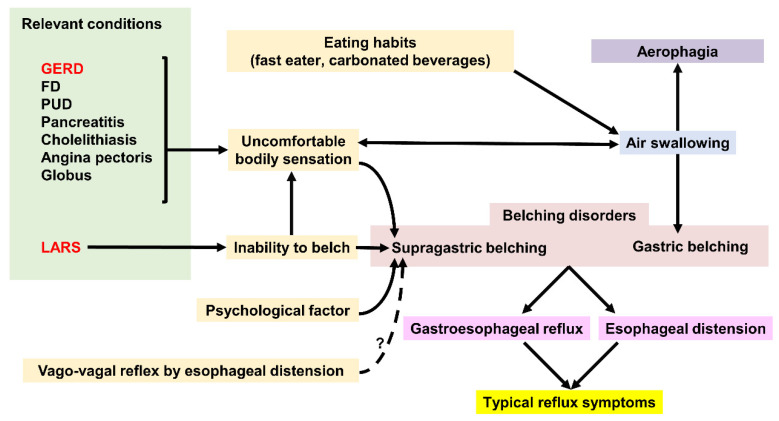
Factors affecting Gastric belching and Supragastric belching. GERD; gastroesophageal reflux disease, FD; functional dyspepsia, PUD; peptic ulcer disease, LARS; laparoscopic anti-reflux surgery.

**Table 1 jcm-09-03360-t001:** Prevalence of belching symptom.

Author	Country	Study Subjects	Number of People Surveyed (*n*)	Overall Prevalence of Belching	Prevalence of Belching Symptom without GERD	Prevalence of Belching Symptom with GERD	Definition of GERD
Westbrook et al. [25]	Australia	General population	2300	6.7%			
Bor et al. [26]	Turkey	General population	630	15.9%	11.5% (*n* = 393)	23.2% (*n* = 237)	Heartburn and/or regurgitation
Rey et al. [27]	Spain	General population	2500	20.5%	12.6% (*n* = 1709)	37.5% (*n* = 791)	Heartburn and/or regurgitation
Li et al. [28]	China	Outpatients	15283	28.8% (*n* = 13282)	26.3% (*n* = 12257)	58.5% (*n* = 1025)	RDQ score > 12
Kessing et al. [14]	Netherland	GERD	90	-	-	75.6%	Reflux symptoms
Klauser et al. [29]	Germany	GERD	304			44.7%	Esophageal symptom
Dore et al. [30]	Italy	GERD	266	-	-	26.3%	Heartburn, regurgitation, dysphagia or odynophagia
Yarandi et al. [31]	Iran	GERD	1522	-	-	4.1%	Heartburn, regurgitation and dysphagia
Ribolsi et al. [32]	Italy	GERD	573	-	-	62.7%	Heartburn, regurgitation, or noncardiac chest pain
Lin et al. [33]	USA	GERD	180			70%	Positive DeMeester score (>14.2), endoscopic esophagitis, or Barrett’s esophagus

RDQ; Reflux Disease Questionnaire.

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
