# Peer review of "Belching in Gastroesophageal Reflux Disease: Literature Review"

_jcm, 2020, doi:10.3390/jcm9103360_

Round 1
Reviewer 1 Report
Overall: Nicely written review of a topic that is confusing for a lot of practitioners, including gastroenterologists! Well done presenting/reviewing the concepts. Minor grammatical/word-smith issues described below. A figure showing manometry features of supragastric belching (e.g. repeated episodes of decreased/increased esophageal pressure) might be helpful. Also additional ph/impedance figures showing relationship of SGB to reflux would be helpful.
Abstract “
Grammar improvement needed: “Belching is a very common phenomenon, however, becomes bothersome if excessive.”
“ the latter is a behavior disorder “ should be behavioral
Introduction
“ SGB has to be diagnosed appropriately as requiring dedicated psychological therapy” should be “as it requires”
“ from the brainstem releases nitric oxide and vasoactive intestinal peptide, which relaxes LES” should be “the les”
Two types of belching: Gastric belching and Supragastric belching (Figure 1)
Line 64 “to ease such unpleasant sensation in some way,..” should be “such an unpleasant”
Line 65 “sometime encounter patients remembering the time when suddenly commencing SGB excessively” needs rewording … “we sometimes encounter patients who remember the exact moment or time when they began having excessive SGB” or something like that.
Line 66 “In such case” should be “in such cases”
Figure 1 A – do you think it would be helpful to show both arrows – ie an arrow reflecting liquid reflux
Conceptual change of Belching disorders and Aerophagia in Rome diagnostic criteria (Figure 2)
Lines 100-102 needs rewording: “ However, the two disorders are segregated from each other recently because they show the discrete characteristics as for esophageal peristalsis, main symptom and the location of swallowed air.” Should be something like “they have distinctive features or physiology” or something like that. Sentence as written is confusing.
Epidemiology (Table 1)
Line 114 “in general population” should be “in the …”
Table 1 – are some numbers missing? E.g. Klauser et al “ showed no difference in the prevalence of belching between normal and 118 pathological acid exposure in GERD” … what was the prevalence of belching without GERD – assuming that’s what normal refers to no GERD?
Belching and Gastroesophageal reflux disease (Figure 3)
Line 132 “PPI refractory GERD have much prandial” should include “more”
Line 138 “SGB can induce reflux, of which mechanism remain unclear” should be, “the mechanism of which is unclear”’
Lines 138-139 - might benefit from some slight wordsmithing CAPITAL LETTERS INSERTED “There are three possibilities as follows: 1) SGB-induced esophageal distension might provoke A TLESR [37,38], 2) AN INCREASE IN abdominal pressure might PROPEL the gastric contentS through the LES, 3) TLESR might elicit a reflex of UES relaxation which IN TURN letS the air into the esophagus”
Line 143 “This study also proved …” should be “…showed…”
Line 144 “not coincidentally occurs close to reflux.” Should be “not occur coincidentally close to reflux”
Impact of Supragastric belching (SGB) on PPI-refractory GERD 145
“Recent studies have increasingly raised awareness of potential .. Our study [2] found the prevalence varies between reflux phenotypes as 37.7% in non-erosive reflux disease (NERD), 39.7% in reflux hypersensitivity (RH) and 22% in functional heartburn (FH).” What is evidence for SGB erosive esophagitis; asking because later in the review you invoke ARS as a treatment
Line 149 “varies between reflux phenotypes as ..” The word As should be “with”
Line 152 – did you define RH before?
Line 158 “as Japanese have lower relevance of SGB 158 to reflux symptoms compared to British” needs rewording
Line 162-168 needs some editing; concept is good but needs editing
Clinical approach and Treatment for Belching
Line 225: “Therefore, clinicians have to pay attention to impedance tracing so that hidden SGB would not be overlooked. As approximately 50%-60% of the SGBs exists close to reflux episodes [2,6], manual editing of automatic analysis of all reflux episodes can identify pathologic SGB” Perhaps it Might be helpful to have an impedance tracing here – similar to Figure 1B with a SGB before and or after the liquid reflux episode?
Author Response
We thank reviewer 1 for his/her comments to improve the manuscript.
Reviewer 1:
Overall: Nicely written review of a topic that is confusing for a lot of practitioners, including gastroenterologists! Well done presenting/reviewing the concepts. Minor grammatical/word-smith issues described below. A figure showing manometry features of supragastric belching (e.g. repeated episodes of decreased/increased esophageal pressure) might be helpful. Also additional ph/impedance figures showing relationship of SGB to reflux would be helpful.
- We amended the manuscript as per your comments. Besides, we added some images of GB and SGB in HRM with impedance and other patterns of the relationship between SGB and reflux in impedance-pH monitoring (Page 3,4, Figure 1 and 2).
Abstract “
Grammar improvement needed: “Belching is a very common phenomenon, however, becomes bothersome if excessive.”
- We changed the sentence as follows: Belching is a common phenomenon. However, it becomes bothersome if excessive. (Page 1, line 10)
“ the latter is a behavior disorder “ should be behavioral
- We amended it as per your comment. (Page 1, line 13)
Introduction
“ SGB has to be diagnosed appropriately as requiring dedicated psychological therapy” should be “as it requires”
- We amended it as per your comment. (Page 1, line 36)
“ from the brainstem releases nitric oxide and vasoactive intestinal peptide, which relaxes LES” should be “the les”
- We amended it as per your comment. (Page 2, line 46)
Two types of belching: Gastric belching and Supragastric belching (Figure 1)
Line 64 “to ease such unpleasant sensation in some way,..” should be “such an unpleasant”
- We amended it as per your comment. (Page 2, line 64)
Line 65 “sometime encounter patients remembering the time when suddenly commencing SGB excessively” needs rewording … “we sometimes encounter patients who remember the exact moment or time when they began having excessive SGB” or something like that.
- We amended it as per your comment. (Page 2, line 64-66)
Line 66 “In such case” should be “in such cases”
- We amended it as per your comment. (Page 2, line 66)
Figure 1 A – do you think it would be helpful to show both arrows – ie an arrow reflecting liquid reflux
- We think that would be helpful. Separate arrows were added to indicate liquid reflux in Figure 1.
Conceptual change of Belching disorders and Aerophagia in Rome diagnostic criteria (Figure 2)
Lines 100-102 needs rewording: “ However, the two disorders are segregated from each other recently because they show the discrete characteristics as for esophageal peristalsis, main symptom and the location of swallowed air.” Should be something like “they have distinctive features or physiology” or something like that. Sentence as written is confusing.
- We amended it as per your comment. (Page 3, line 118)
Epidemiology (Table 1)
Line 114 “in general population” should be “in the …”
- We amended it as per your comment. (Page 5, line 130)
Table 1 – are some numbers missing? E.g. Klauser et al “ showed no difference in the prevalence of belching between normal and 118 pathological acid exposure in GERD” … what was the prevalence of belching without GERD – assuming that’s what normal refers to no GERD?
- In the table, non-GERD means subject without reflux symptoms. In a study by Klauser et al., all the patients had reflux symptoms irrespective of the extent of esophageal acid exposure. Therefore, it is difficult to calculate the prevalence of belching symptom of normal subjects from the study.
Belching and Gastroesophageal reflux disease (Figure 3)
Line 132 “PPI refractory GERD have much prandial” should include “more”
- We amended it as per your comment. (Page 6, line 148)
Line 138 “SGB can induce reflux, of which mechanism remain unclear” should be, “the mechanism of which is unclear”’
- We amended it as per your comment. (Page 6, line 154)
Lines 138-139 - might benefit from some slight wordsmithing CAPITAL LETTERS INSERTED “There are three possibilities as follows: 1) SGB-induced esophageal distension might provoke A TLESR [37,38], 2) AN INCREASE IN abdominal pressure might PROPEL the gastric contentS through the LES, 3) TLESR might elicit a reflex of UES relaxation which IN TURN letS the air into the esophagus”
- Thank you for the advice. We amended it as per your comment. (Page 6, line 155-157)
Line 143 “This study also proved …” should be “…showed…”
- We amended it as per your comment. (Page 6, line 159)
Line 144 “not coincidentally occurs close to reflux.” Should be “not occur coincidentally close to reflux”
- We amended it as per your comment. (Page 6, line 160)
Impact of Supragastric belching (SGB) on PPI-refractory GERD 145
“Recent studies have increasingly raised awareness of potential .. Our study [2] found the prevalence varies between reflux phenotypes as 37.7% in non-erosive reflux disease (NERD), 39.7% in reflux hypersensitivity (RH) and 22% in functional heartburn (FH).” What is evidence for SGB erosive esophagitis; asking because later in the review you invoke ARS as a treatment
- We are afraid that we do not have data about the prevalence of SGB in erosive esophagitis.
Line 149 “varies between reflux phenotypes as ..” The word As should be “with”
- We amended it as per your comment. (Page 7, line 165)
Line 152 – did you define RH before?
- No, we did not. We added the definition of NERD, RH and FH to the manuscript. (Page 7, line 166-168).
Line 158 “as Japanese have lower relevance of SGB 158 to reflux symptoms compared to British” needs rewording
- We amended the sentence as follows: “SGB is less associated with reflux symptoms in Japanese than in British.” (Page 7, line 176-177)
Line 162-168 needs some editing; concept is good but needs editing
- We modified the paragraphs to make it easier to understand as below. (Page 7, line 181-190)
“Laparoscopic Nissen fundoplication (LNF) is one of the most common procedure for GERD, which creates a 360° wrap of the stomach around the esophagogastric junction (EGJ) and repair the crura if necessary. By tightening the EGJ, LNF eliminates reflux as well as GBs. However, LNF makes patients feel difficult to belch and can produce other gas-related symptoms including bloating or flatulence [41-43]. These discomforts presumably lead the patients to perform SGB as a futile attempt to belch, and end up increasing the number of SGBs postoperatively [44].
Regarding laparoscopic partial fundoplication (LPF) which constructs a looser wrap (e.g. 270° posterior wrap (Toupet)), it can reduce gastroesophageal reflux similar to LNF and lessen postoperative gas-related symptoms remaining the ability to belch. However, LPF also increases the number of SGBs [45,46].”
Clinical approach and Treatment for Belching
Line 225: “Therefore, clinicians have to pay attention to impedance tracing so that hidden SGB would not be overlooked. As approximately 50%-60% of the SGBs exists close to reflux episodes [2,6], manual editing of automatic analysis of all reflux episodes can identify pathologic SGB” Perhaps it Might be helpful to have an impedance tracing here – similar to Figure 1B with a SGB before and or after the liquid reflux episode?
- Thank you for the advice. We added those images into Figure 1.
Reviewer 2 Report
Very nice and important review.
- Some English language errors - please review your manuscript again
- Table 1 is not in order, please check and correct
- Please add more pH-impedance and manometry & impedance monitoring tracings that show the differences between GB vs. SGB vs. Normal
Author Response
We thank reviewer 2 to provide us useful comments to improve the manuscript.
Reviewer 2:
1. Some English language errors - please review your manuscript again
- We amended the manuscript grammatically following the comments of reviewer 1.
2. Table 1 is not in order, please check and correct
- We sorted the studies in the order of reference numbers.
3. Please add more pH-impedance and manometry & impedance monitoring tracings that show the differences between GB vs. SGB vs. Normal
- We added more images about GB and SGB in impedance-pH monitoring and HRM combined with impedance (Figure 1 and 2).